# Assessment of the Effectiveness of Obstructive Sleep Apnea Treatment Using Optical Coherence Tomography to Evaluate Retinal Findings

**DOI:** 10.3390/jcm11030815

**Published:** 2022-02-03

**Authors:** Gloria Tejero-Garcés, Francisco J. Ascaso, Paula Casas, Maria I. Adiego, Peter Baptista, Carlos O’Connor-Reina, Eugenio Vicente, Guillermo Plaza

**Affiliations:** 1Department of Otolaryngology, Hospital Clínico Universitario Lozano Blesa, 50009 Zaragoza, Spain; dra_tejero_garces@hotmail.com; 2Department of Ophthalmology, Lozano Blesa University Clinic Hospital, 50009 Zaragoza, Spain; jascaso@gmail.com (F.J.A.); paulacasaspascual@hotmail.com (P.C.); 3Aragón Health Research Institute (IIS Aragon), 50009 Zaragoza, Spain; 4Department of Otolaryngology, Hospital Universitario Miguel Servet, 50009 Zaragoza, Spain; isa.adiego@gmail.com (M.I.A.); eavicenteg@gmail.com (E.V.); 5Department of Otolaryngology, Clínica Universitaria de Navarra, 31008 Pamplona, Spain; peterbaptista@gmail.com; 6Department of Otolaryngology, Hospital Quironsalud Marbella, 29603 Marbella, Spain; coconnor@us.es; 7Department of Otolaryngology, Hospital Universitario de Fuenlabrada, Universidad Rey Juan Carlos, 28942 Madrid, Spain; 8Department of Otolaryngology, Hospital Universitario Sanitas La Zarzuela, 28942 Madrid, Spain

**Keywords:** retina, retinal nerve fiber layer, obstructive sleep apnea syndrome, optical coherence tomography, OCT, CPAP, upper airway surgery

## Abstract

Retinal findings may change in patients with obstructive sleep apnea syndrome (OSAS). The present study aims to evaluate several retinal findings, such as macula layer thickness, the peripapillary retinal nerve fiber layer, and the optic nerve head in patients with OSAS, using optical coherence tomography (OCT); it also aims to monitor the result of several types of treatment of OSAS with OCT. A prospective comparative study was designed. Patients were recruited at a Sleep Unit of a University Hospital and underwent comprehensive ophthalmological examinations. Following exclusion criteria, fifty-two patients with OSAS were finally included. Patients were examined by OCT twice: once before treatment, and again after six months of treatment. In mild–moderate patients, where retinal swelling had been demonstrated, retinal thicknesses decreased [fovea (*p* = 0.026), as did inner ring macula (*p* = 0.007), outer ring macula (*p* = 0.015), and macular volume (*p* = 0.015)]. In severe patients, where retinal atrophy had been observed, retinal thickness increased [fovea (*p* < 0.001)]. No statistically significant differences in efficacy between treatments were demonstrated. In conclusion, OCT can evaluate the retina in patients with OSAS and help to monitor results after treatment. In severe OSAS, retinal thickness increased six months after treatment.

## 1. Introduction

Obstructive sleep apnea syndrome (OSAS) is one of the most frequent sleep disorders in our population [1]. A recent estimation has shown that 13% of middle-aged men and 5% of middle-aged women are affected. An increase in prevalence by 30% between 1990 and 2010 was seen [2]. Continuous and maintained oxygen desaturation during sleep results in metabolic, cardiovascular, and neuropsychiatric consequences that impact quality of life and increase mortality [1,3]. Cognitive alterations have, in particular, been associated with both the reduction in the overall performance and the memory abilities of patients with OSAS [4].

Intermittent hypoxia enhances a sympathetic response and oxidative stress that result in systemic and local inflammation [5]. Intermittent hypoxemia and sleep fragmentation cause continuous sympathetic stimulation with increased cardiac arrhythmogenic risk [6]. Intermittent hypoxemia disorder is also correlated with degeneration of the gray matter of the central nervous system, with increased risk for autonomic pathologies and systemic inflammation [7].

This high incidence and severe comorbidity make an early diagnosis essential [1,8].

Repeated sleep fragmentation causes significant oxidative stress with a chronic systemic inflammatory state. Numerous biomarkers have been proposed in this regard in the literature. CPAP (continuous positive airway pressure) or surgical treatment, however, are demonstrated to reduce inflammation biomarkers [9].

Currently, sleep studies, such as polysomnography and cardio–respiratory polygraphy, allow the diagnosis and follow-up treatment of OSAS [1,8]. However, these techniques are expensive and not widely available, generating high demand and long waiting lists.

The most accepted marker for this disease’s severity is the sleep studies apnea/hypopnea index (AHI). However, AHI only shows the number of times a patient presents respiratory events, but does not measure its effect on the body, which can vary between patients with a similar AHI [1,8,10,11]. Respiratory indices such as AHI have recently been associated with the severity of nasal disorders, especially in chronic rhinitic inflammatory states or nasal vascular hyperactivity [12].

The retina, a part of the central nervous system, may express changes during chronic and intermittent hypoxia in OSAS patients. Optical coherence tomography (OCT) has been used in various neurological disorders to study the retina [13,14,15,16]. In addition, it provides fast and non-invasive information on macula layer thickness, the peripapillary retinal nerve fiber layer (RNFL) and the optic nerve head (ONH) [17]. Some studies have already demonstrated retinal changes in OSAS patients, such as increased thickness in mild-moderate OSAS patients, a so-called inflammation, or a decrease in retinal thickness—a degenerative neurological process—in severe OSAS patients [18,19,20,21,22,23,24,25,26,27,28,29,30,31,32].

Furthermore, OCT has recently been used to monitor the effect of CPAP or surgical treatment for OSAS, showing how OCT is useful for checking improvements in the retina and optic nerve measurements [33,34,35,36]. These investigations have shown choroidal and retinal improvement after treatment. A recent trial demonstrated that CPAP treatment could improve visual sensitivity and increase retinal thickness in patients with OSA [34].

In the present study, we aimed to evaluate OCT as a tool to monitor the response to therapy through retinal changes such as macula thickness, RNFL, and ONH, after CPAP therapy and after surgical treatment, in two related groups of patients with OSAS; we then aimed to review the current role of OCT as a monitoring tool after OSAS treatment through the literature.

## 2. Materials and Methods

This research was approved by the Institutional Review Board and Ethics Committee of a University Hospital and planned as a prospective non-randomized comparative study. Informed consent was obtained from all participants. The present study included patients older than 18 years, presented to the Sleep Disorders Unit of Hospital Universitario Miguel Servet (Zaragoza, Spain). All of the patients complained of snoring and witnessed apnea episodes or excessive daytime sleepiness for over two years. A complete otolaryngologic exam was performed, including nasopharyngolaryngoscopy, and an ophthalmological examination was performed with OCT measurements.

Each subject underwent an attended overnight cardio–respiratory polygraphy (Bitmed^®^ NGX320 Sibel, Barcelona, Spain) conducted by the Pulmonology Department at the same University Hospital. The apnea/hypopnea index (AHI) was calculated as the total number of apneas and hypopneas per hour of electroencephalographic sleep, following AASM criteria. OSAS severity was classified as mild for an AHI between 5 and 15, moderate for an AHI greater than 15 and less than 30, and severe for an AHI greater than 30. All studies were scored and read by a physician unaware of the study’s aim and, therefore, masked to patients’ ophthalmic evaluation results.

The ophthalmologic evaluation included best-corrected visual acuity (BCVA), slit-lamp biomicroscopy, Perkins applanation tonometry, gonioscopy with a four-mirror contact lens, and fundoscopy. OCT was performed with the time-domain optical coherence tomography (TD-OCT) Stratus of Carl Zeiss Meditec Inc., Dublin, CA, USA, following 1% tropicamide instillation to dilate the pupils [5]. Only high-quality images (signal strength ≥ 7) were included [5,20]. Each patient underwent scans to measure macular parameters (foveal thickness, inner ring thickness, outer ring thickness and macular volume), peripapillary RNFL parameters (average thickness and superior, inferior, temporal and nasal quadrant thicknesses) and ONH morphometric parameters (vertical integrated rim area (VIRA), horizontal integrated rim width (HIRW), disc area, cup area, rim area, and cup/disc ratios) (Figure 1), all during the same visit.

Both eyes of each patient were included, except when an exclusion criteria appeared in one eye, assuming that the influence of OSAS could be asymmetric, following Shrier et al. [37].

Exclusion criteria included: patients who suffered from ocular disease (Humphrey visual field defects compatible with glaucoma or an intraocular pressure (IOP) > 21 mmHg; refractive error over 4 spherical diopters and 3 diopters of astigmatism; posterior segment pathology or media opacification; maculopathy; ocular surgery; or trauma); patients who suffered from systemic disorders (uncontrolled systemic pathology, such as diabetes mellitus or arterial hypertension, or mental, neurodegenerative or chronic inflammatory disorders) which could damage their retina or could prevent an appropriate study; and patients who would only be treated with dietetic and hygienic measures, or had been previously treated with CPAP or surgery.

Treatment that consisted of CPAP therapy, surgery, or a combination of both was given once patients had completed the OCT examination. Each treatment was specifically chosen according to the patient’s AHI, comorbidity, physical examination, body mass index (BMI), and preference, following current guidelines [5,38,39].

A well-trained technician conducted individual training and the optimal pressure of CPAP therapy, as recommended [1]. Patients were encouraged to sleep with the PAP therapy to reach adequate adherence. Therapy adherence was defined as using PAP therapy for an average of four hours a night for at least 70% of the nights. Only patients with PAP adherence were included for comparing outcomes.

Patients were selected according to ENT and fibroscopy examination, Muller’s maneuver, and the severity of OSAS disease according to PSG parameters for the surgical group. In all cases, septoplasty and palatopharyngoplasty were performed, and multilevel surgery was performed if required. CPAP was also added for those cases that did not improve after surgery, that is, having an AHI > 15 three months after surgery.

After a minimum of six months of treatment, all of the patients underwent final cardio–respiratory polygraphy and a complete bilateral ophthalmological examination. 

Data analysis was conducted using SPSS software version 22.0 (SPSS, Inc., Chicago, IL, USA). Values are presented as mean ± standard deviation (SD) and percentages, expressed in microns (μm) for the peripapillary RNFL thickness and macular retinal thickness, and in mm^3^ for macular volume. Once normal distribution and homogeneity were confirmed with Kolmogorov–Smirnov and Levene’s tests, respectively, a comparison between means was carried out using a paired *t*-test and ANOVA test, through the General Linear Model (GLM) procedure for more than two groups.

A paired *t*-test was used to compare initial and control OCT data, constantly comparing the same eye side from baseline to control measures. The correlation between AHI and significant ophthalmologic variables was evaluated using Pearson’s linear correlation coefficient. A *p*-value < 0.05 was considered statistically significant. The sample size was defined in 92 eyes to be evaluated, with an error range of 3% for a confidence level of 95% under the assumption of maximum variance (*p* = q = 0.5).

Finally, a systematic review of published data, aiming to identify the published studies on OCT and OSAS, was conducted according to STROBE guidelines [40].

## 3. Results

Sixty patients (120 eyes) were initially recruited. Six patients refused to continue at some point during the study, leaving 54 patients to be analyzed. Of a total of 108 eyes that were examined, ten eyes of ten different patients had to be excluded due to previous ophthalmological pathologies. Finally, 98 eyes from 52 patients were included in the study (Figure 2 with flow diagram).

Among our patients, 42 (80.7%) were male, and the mean age was 50.1 ± 12.6 years. According to the severity of OSAS, four patients (7.7%) had mild OSAS (AHI ≥ 5), 19 (36.6%) had moderate OSAS (AHI ≥ 15) and 29 (55.7%) had severe OSAS (AHI ≥ 30). They were grouped into mild–moderate (23 patients) or severe (29 patients) cases for statistical purposes.

Moreover, 32.7% had controlled arterial hypertension, 7.7% had controlled diabetes mellitus, 28.8% were smokers of < 5 packs/year, and 26.8% former smokers. 13.5% had a normal BMI, 44.2% were overweight, and 42.3% had obesity. The mean BMI was 28.8 in mild–moderate OSAS, and 29.8 in severe OSAS.

When describing the three groups of treatment, 17 (32.7%) of our patients were treated with CPAP, 15 (28.8%) with surgery, and 20 (38.5%) underwent a combination of CPAP and surgery. The initial AHI was significantly lower in those cases only treated with surgery (27.1 ± 15.3), as compared to those treated with CPAP (46.5 ± 20.4) or a combination of CPAP and surgery (54.1 ± 29.3). Furthermore, the initial AHI was significantly worse in severe OSAS cases that were treated with a combination of CPAP and surgery than those only treated with surgery (69.7 vs. 43.2; Table 1).

A significant improvement in AHI was observed in the severe OSAS group following surgery (43.2 ± 14.6 to 11.2 ± 8.0) and a combination of CPAP and surgery (69.7 ± 25.6 to 38.2 ± 23.4; Table 1).

At baseline, the retinal parameters in OCT varied according to the OSA’s severity, with statistically significant differences. Regarding the macular parameters, the inner ring thickness (*p* = 0.03) and the macular volume (*p* = 0.04) were lower in severe OSAS than in mild–moderate OSAS. Concerning the RNFL parameters, the total average thickness (*p* = 0.003), superior quadrant thickness (*p* = 0.04), inferior quadrant thickness (*p* = 0.01), and temporal quadrant thickness (*p* = 0.008) were all lower in severe OSAS. Finally, concerning the optic nerve head morphological parameters, HIRW was higher in mild–moderate OSAS (*p* = 0.01), whereas the excavation area (cup area) was more significant in severe OSAS (*p* = 0.02) (Table 2).

In the mild–moderate patient OSAS group, where retinal swelling was demonstrated, all of the macular measurements in the OCT showed a statistically significant decrease after six months of treatment (Table 3 and Figure 3).

Every RNFL parameter decreased, but not in a statistically significant way (see table in Appendix A), and none of the ONH morphometric parameters showed changes following treatment.

In the severe OSAS group, where retinal atrophy had been observed, thickness increased, and all of the macular parameters increased their value. However, only the foveal thickness increase was statistically significant (Table 4 and Figure 4). All RNFL thicknesses increased their value. Still, only the average thickness increased in a statistically significant way (see table in Appendix A, and Figure 4), and finally, no changes were observed in morphometric ONH parameters.

In summary, analyzing OCT changes after CPAP and/or surgery, we found a substantial improvement in foveal thickness and RNFL average thicknesses in both groups. Still, we could not see a significant difference between them. Lastly, we calculated the correlation between AHI and OCT, obtaining no relationship between the changes in the retinal parameters and changes in the AHI after six months of treatment. 

## 4. Discussion

Intermittent airway obstruction in OSAS leads to hypoxia, hypercapnia, and changes in intrathoracic pressure. Consequently, perfusion alterations due to autonomic, hemodynamic, humoral, and neuroendocrine changes may affect the brain of OSAS patients [13,14,15,16]. A straightforward way of studying the neurological consequences of this entity could be reached using information about the retinal structure in OSAS patients [30,41]. 

OCT is a technique that provides real measurements of the macula, peripapillary retinal layers, and the optic nerve, allowing a fast, non-invasive way to show possible retinal changes in OSAS patients. Accordingly, retinal changes demonstrated by OCT could function as a window into the brain and can be used as a biomarker [17,41]. 

As previously reported [18,19,20,21,22,23,24,25,26,27,28,29,30], during sleep, intermittent apneic episodes with oxygen desaturation activate proinflammatory and procoagulant mechanisms accompanied by the adrenergic system. These changes promote endothelial dysfunction and oxidative stress with an increase in vascular resistance that, in the end, compromises the optic nerve perfusion and leads to a reduction in retinal thicknesses. These findings can easily be shown with OCT.

A meta-analysis presented by Yu et al. reviewing 10 case-control studies, and another one by Wang et al. that includes 17 studies on OCT in OSAS, show that the average RNFL thickness in OSAS patients is significantly reduced compared to healthy controls [27,28]. Following our systematic review, we discovered how several authors have proposed OCT as a valuable tool to monitor and assess the severity of OSAS in patients, as can be seen in Table 5.

Lin et al. [19], Sagiv et al. [24], and Ngoo et al. [43] observed a decrease in the average peripapillary RNFL thickness and the superior, inferior, and temporal quadrant thicknesses in OSAS patients compared to those in healthy subjects. On the contrary, Casas et al. [20] and Guven et al. [44] found a decrease in the RNFL thickness just in the nasal quadrant in OSAS patients, whereas Adam et al. [21] and Kücük et al. [42] did not find any significant difference in RNFL thickness between OSAS patients and healthy controls. In the present study, we observed that baseline macular thicknesses and average RNFL thicknesses were significantly lower in severe OSAS than in mild–moderate OSAS, as found by Yu et al. [29]. Our findings in severe cases may result from the long-term oxidative process in OSAS, as our patients were recruited after having apneas for more than two years, compared to less severe cases. However, differences in retinal thickness were never so great as to jump several steps from the percentile of the normal distribution considered by Stratus (i.e., red sector to green sector). 

Regarding the optic nerve parameters in OCT, Lin et al. [19], Casas et al. [20], and Uslu et al. [31] reported a greater excavation optic nerve head area (cup area) in OSAS patients than in controls. Kücük et al. [36] showed that the lamina cribosa thickness was significantly thinner in OSAS patients than in controls. On the contrary, Moyal et al. [32] could not find significant differences between OSAS patients and controls. In this present study, baseline OCT results were compatible with the theory of neurovascular damage caused by hypoxia: swelling was observed in the retina of mild–moderate OSAS patients, as opposed to atrophy and more significant excavation in severe OSAS patients.

Recently, OCT has been used to monitor the effect of CPAP or surgical treatment for OSAS, showing its utility to check improvements in retina and optic nerve measurements [33,34,35,36]. Zengin et al. [33] studied 44 OSAS patients treated with CPAP, who were followed during a whole year with OCT examinations every three months, and compared those results to those of healthy subjects. Baseline OCT data showed no differences between both samples; however, following one year of CPAP therapy, a lower average peripapillary RNFL, and nasal, inferior, and superior quadrant thicknesses were described in the patients with OSAS group as compared to the control group. They also studied the correlation between the AHI and the RNFL thickness, observing a weak negative correlation.

Similarly, Lin et al. [34] presented a prospective study on 32 OSAS patients treated with CPAP who underwent an OCT three months after treatment. They found that the inferior quadrant and nasal-inferior sector of the RNFL thickness significantly improved after treatment. In addition, the macula layer thickness in the superior-inner sector, inferior-outer sector, nasal-outer sector, superior hemisphere, and inferior hemisphere was also significantly improved after treatment. The improvement of macular layer thickness in the superior-inner sector positively correlated with the AHI and desaturation index improvements. Naranjo-Bonilla et al. [45] have recently reported a prospective study including 28 patients treated with CPAP and 12 untreated, again showing normalization of the choroidal thickness measured by OCT in treated patients.

Lin et al. [35] presented a prospective study on 108 OSAS patients, treated with upper airway surgery, who were monitored by polysomnographic findings and OCT. The thickness of the macular layer in the nasal-outer, superior-inner, temporal-inner, inferior-inner and nasal-inner sectors, and the total macular layer thickness, significantly increased 6 months after upper airway surgery in the severe OSAS group. On the contrary, Kaya et al. [36] presented a prospective study on 34 OSAS patients treated with expansion sphincter pharyngoplasty. After six months, the preoperative and postoperative AHI scores and average oxygen saturation values were significantly different, but there was no significant difference between the preoperative and postoperative RNFL thicknesses.

Jayakumar et al. [46] published a prospective study including 36 patients, comparing CPAP, uvulopalatopharyngoplasty, and no treatment. They showed that choroidal thickness and vascularity improved after six months of both surgery and CPAP. Our study also found an improvement in OCT findings in severe OSAS patients after CPAP or surgical treatment. However, we did not obtain any correlation between changes in the AHI and changes in the OCT after surgical treatment. The most relevant finding in our study is that the foveal thickness and retinal nerve fibers’ (RNFL) average thickness improved after six months of treatment in severe cases. That is, treatment can improve the retinal consequences of OSAS. However, we could not find improvements in optic nerve parameters. The findings we observed, together with the systemic involvement of OSAS, reinforce the idea of the need for early diagnosis and treatment to avoid possible complications and alterations, such as those found in the optic nerve in present study.

One of the most remarkable findings of the present study is the inverse evolution with respect to foveal thickness and RNFL that we found in patients with severe OSAS. A possible hypothesis to this finding could be the major alteration of the choroidal perfusion in the most severe cases, which would generate a greater vasoconstriction and a decreased thickness in the retinal overlying layers. Therefore, this finding suggests that in the earliest stages of the disease, inflammatory phenomena would predominate, with increased tissue volume. Nevertheless, in the most severe stages, the vasoconstriction and vascular remodeling would dominate, generating a surrounding tissue atrophy [47].

As recently reviewed by D’Souza & Kappor [48], OSAS demonstrated a significant association with retinal vascular disorders. It is associated with the development of diabetic retinopathy, retinal vein occlusion, and central serous chorioretinopathy. Retinal changes may be the first clinical manifestation of otherwise undiagnosed OSA, so it is important to refer patients with new-onset retinal vascular disease for appropriate sleep testing. This reinforces our findings on OCT as a monitor of OSAS.

Lastly, this study has some limitations. First, the research being a single-center study, and having a small sample size, could be considered the main limitation of the present study. This was not a random trial; however, well-matched groups of patients were studied and provided data on the 6-month follow-up of OSAS treatment effects. We accepted a possible asymmetric affection of the CNS by OSAS, ignoring an interocular correlation in the same patient [49]. However, our baseline results did not show such deviation between sides. An apparently common error in statistical analysis of ophthalmic data is to perform statistical tests that do not account for the correlation generally present between observations made for the right and left eyes of a subject [50]. In practice, the routine use of two-eye analyses is strongly recommended. Experience has shown that for ophthalmic data, right- and left-eye observations usually have a high positive correlation [51]. This is because subject characteristics are manifested in both right- and left-eye measurements. Furthermore, there is little penalty for the use of two-eye analysis, even when the inter-eye correlation is zero. In this circumstance, a two-eye analysis produces a slightly conservative estimate of precision [52]. In fact, most ocular complications of OSAS are usually symmetric, except keratoconus [53]. Additionally, a more extended follow-up period could have given us more significant results; nevertheless, both were enough to provide clinically relevant outcomes in our study, and these findings should be supported by large-scale studies.

In conclusion, the present study demonstrated that OCT is an objective method of diagnosis that can inform us about OSAS patients’ clinical situations in an easy and non-invasive way by studying the retinal structure. As recent reviews also confirm [54,55], OCT could be useful for undertaking the follow-up of the treatment of this disorder, showing retinal changes that help to monitor endothelial dysfunction in OSAS, such as foveal thickness and the average thickness of retinal nerve fibers. In severe OSAS, retina thickness increased six months after treatment. For these reasons, a complete ophthalmic examination should be performed at every follow-up for OSAS patients.

## Figures and Tables

**Figure 1 jcm-11-00815-f001:**
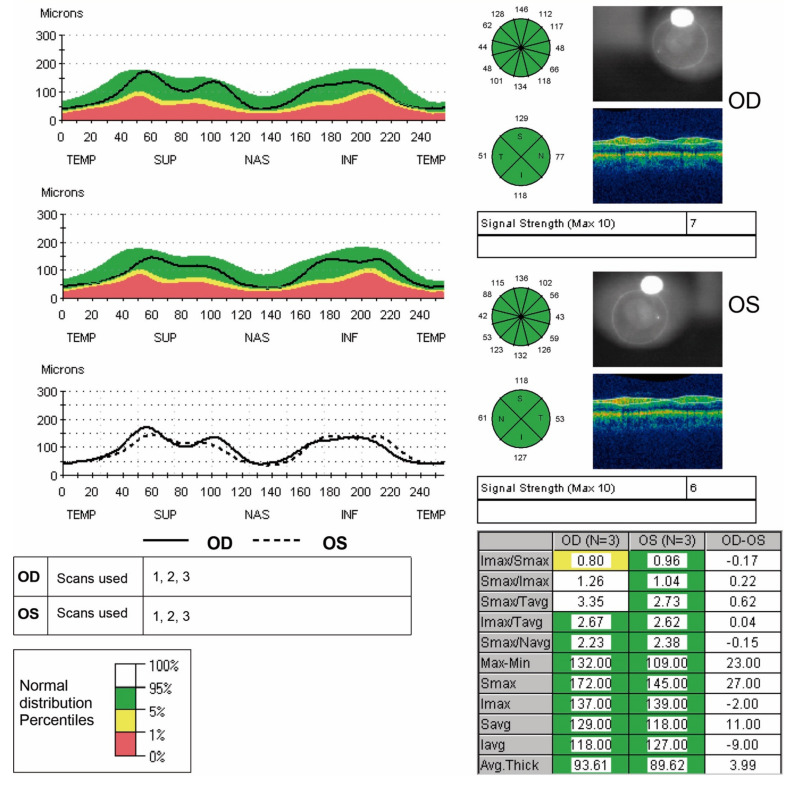
Macular measures with Stratus OCT: From left to right: Retinal thickness/volume; Average thickness (mm); Thickness and volume values; Normative data to compare.

**Figure 2 jcm-11-00815-f002:**
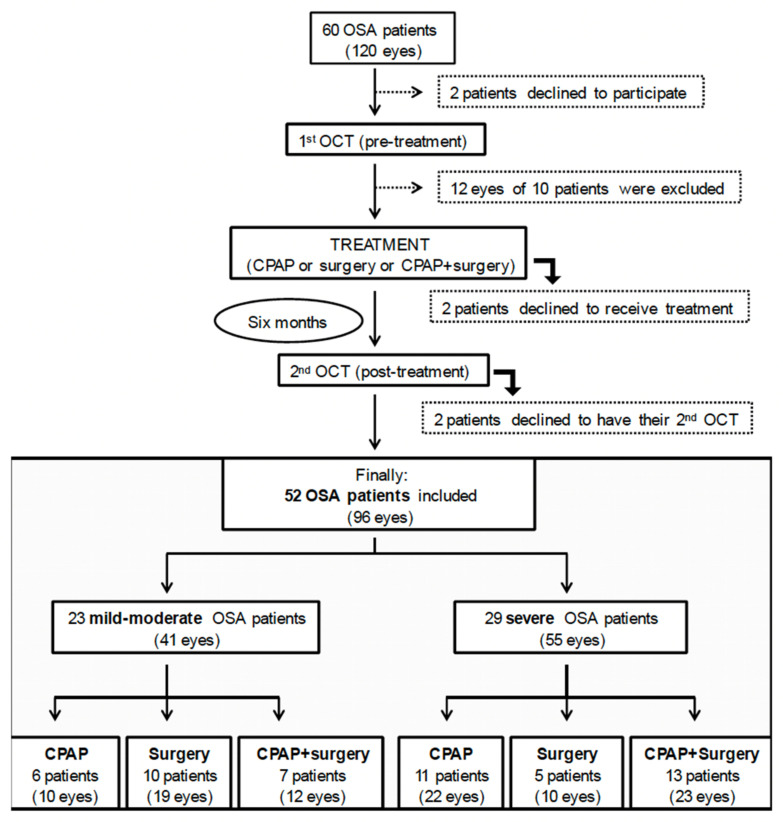
Flow diagram of patients in the study. OCT, optical coherence tomography; OSA, obstructive sleep apnea; CPAP (continuous positive airway pressure).

**Figure 3 jcm-11-00815-f003:**
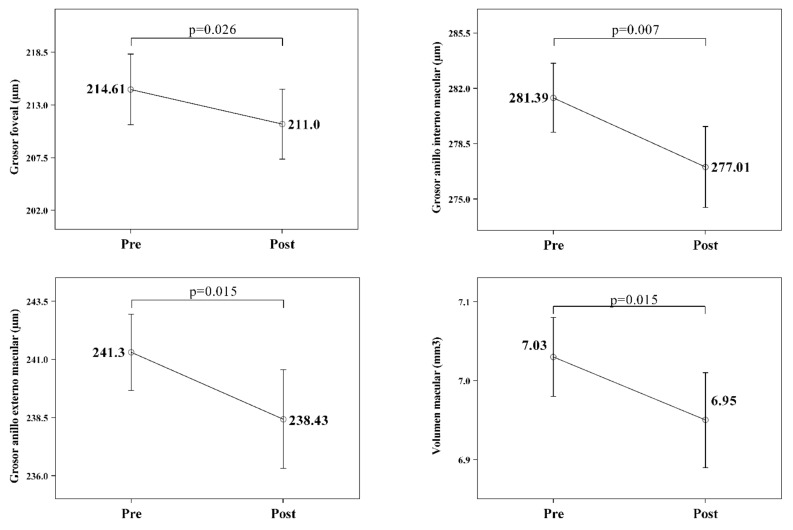
Significant changes in retinal OCT findings after treatment in mild–moderate OSA (43 eyes evaluated from 23 patients), showing a significant decrease in average foveal thickness, inner and outer ring thickness and macular volume.

**Figure 4 jcm-11-00815-f004:**
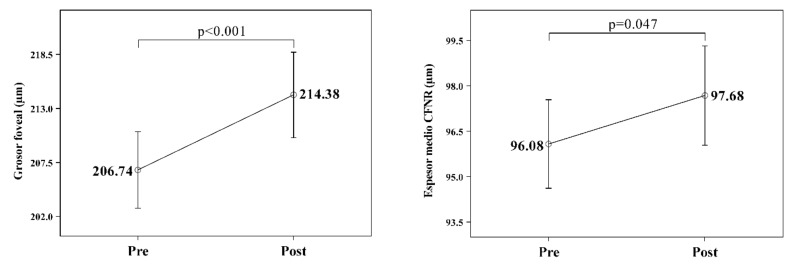
Significant changes in retinal OCT findings after treatment in severe OSA (55 eyes evaluated from 29 patients), showing a significant increase in average foveal and RFNL thickness.

**Table 1 jcm-11-00815-t001:** AHI changes after treatment. GLM: Means ± SD and statistical contrasts of AHI between surgery and combination of CPAP and surgery in both OSA groups (35 patients).

Variable (*n*)	Mean ± SD		
	Pre-Treatment	Post-Treatment	Treatment Effect
			(Comparison pre/post)
			F(d.f.); *p*-Value; (eta^2^)
AHI mild–moderate OSA			F(1.15) = 1.902; *p* = 0.188
Surgery (10)	18.2 ± 7.3	11.6 ± 15.9	(0.113)
CPAP + surgery (7)	24.8 ± 4.8	21.4 ± 13.4	
Total (17)	20.9 ± 7.1	15.6 ± 15.3	
AHI severe OSA			F(1.16) = 31.162; *p* < 0.001
Surgery (5)	43.2 ± 14.6	11.2 ± 8.0	(−0.661)
CPAP + surgery (13)	69.7 ± 25.6	38.2 ± 23.4	
Total (18)	62.4 ± 25.7	30.7 ± 23.6	

GLM: Means ± SD and statistical contrasts of AHI between surgery and combination of CPAP and surgery in both OSA groups (35 patients). GLM—General Linear Model; SD—standard deviation; df—degrees of freedom; eta2—partial eta2 (effect size); *p*—level of statistical significance. AHI, apnea/hypopnea index; OSA, obstructive sleep apnea; CPAP (continuous positive airway pressure).

**Table 2 jcm-11-00815-t002:** Comparison between baseline retinal parameters of both OSA groups (98 eyes).

Variable	Mean ± SD		*p*-Value
	Mild–Moderate OSA (*n* = 41)	Severe OSA (*n* = 55)	
Foveal thickness (µm)	214.6 ± 22.28	206.7 ± 27.38	0.138
Inner ring macular thickness (µm)	281.4 ± 13.4	271.9 ± 15.85	0.003
Outer ring macular thickness (µm)	241.3 ± 1.17	236.4 ± 13.55	0.054
Macular volume (mm^3^)	7 ± 0.33	6.9 ± 0.39	0.041
RNFL average thickness (µm)	102.3 ± 9.06	96.1 ± 1.27	0.003
RNFL superior quadrant thickness (µm)	129.1 ± 12.56	122.3 ± 18.21	0.044
RNFL nasal quadrant thickness (µm)	75.7 ± 16.27	73.3 ± 15.02	0.458
RNFL inferior quadrant thickness (µm)	128.6 ± 16.99	120.1 ± 16.08	0.014
RNFL temporal quadrant thickness(µm)	75.6 ± 11.3	68.7 ± 13.04	0.008
VIRA (mm^3^)	0.8 ± 0.51	0.6 ± 0.31	0.102
HIRW (mm^2^)	2 ± 0.3	1.8 ± 0.28	0.01
Disc area (mm^2^)	2.8 ± 0.69	2.6 ± 0.56	0.115
Cup area (mm^2^)	0.5 ± 0.52	0.9 ± 1.06	0.028
Rim area (mm^2^)	2.1 ± 0.7	1.9 ± 0.81	0.279
Cup/disc area ratio	0.2 ± 0.18	0.3 ± 0.27	0.078
Cup/disc horizontal ratio	0.4 ± 0.19	0.5 ± 0.24	0.123
Cup/disc vertical ratio	0.4 ± 0.17	0.5 ± 0.24	0.108

SD—standard deviation; VIRA—vertical integrated rim area; HIRW—horizontal integrated rim width; RNFL, retinal nerve fiber layer. Cup/disc ratio—excavation/disc ratios; *p*—level of statistical significance.

**Table 3 jcm-11-00815-t003:** Evolution of macular parameters in the mild–moderate OSA group (43 eyes evaluated from 23 patients). GLM—General Linear Model; SD—standard deviation; df—degrees of freedom; eta2—partial eta2 (effect size); *p*—level of statistical significance.

Variable (*n*)	Mean ± SD		
	Pre-Treatment	Post-Treatment	Treatment Effect (Comparison pre/post)
			F(d.f.); *p*-Value; (eta^2^)
Foveal thickness (µm)			F(1.38) = 5.391; *p* = 0.026
CPAP (10)	215.1± 20.3	209.2 ± 18.1	(0.124)
Surgery (19)	215.9 ± 23.9	215.6 ± 20.7	
CPAP + surgery (12)	212.1 ± 22.9	205.2 ± 28.0	
Total	214.6 ± 22.3	211 ± 22.4	
Macular inner ring thickness (µm)			F(1.38) = 8.074; *p* = 0.007
CPAP (10)	284.7 ± 9.7	281.7 ± 12.9	(0.175)
Surgery (19)	281.5 ± 16.0	279.5 ± 16.5	
CPAP + Surgery (12)	278.5 ± 11.7	269.2 ± 16.7	
Total	281.4 ± 13.4	277 ± 16.3	
Macular outer ring thickness (µm)			F(1.38) = 6.485; *p* = 0.015
CPAP (10)	243.8 ± 8.5	240.2 ± 14.6	(0.146)
Surgery (19)	242.2 ± 10.7	241.9 ± 12.5	
CPAP + Surgery (12)	237.8 ± 10.4	231.3 ± 12.7	
Total	241.3 ± 10.2	238.4 ± 13.6	
Macular volume (mm^3^)			F(1.38) = 6.515; *p* = 0.015
CPAP (10)	7.1 ± 0.3	7 ± 0.4	(0.146)
Surgery (19)	7.1 ± 0.4	7.1 ± 0.4	
CPAP + Surgery (12)	6.9 ± 0.3	6.7 ± 0.4	
Total	7 ± 0.3	6.9 ± 0.4	

**Table 4 jcm-11-00815-t004:** Evolution of macular parameters in the severe OSA group (55 eyes evaluated from 29 patients). GLM—General Linear Model; SD—standard deviation; df—degrees of freedom—eta2: partial eta2 (effect size); *p*—level of statistical significance.

Variable (*n*)	Mean ± SD		
	Pre-treatment	Post-treatment	Treatment Effect (comparison pre/post)
			F(f.g.); *p*-Value (eta^2^)
Foveal thickness (µm)			F(1.50) = 16.780; *p* < 0.001
CPAP (22)	215.2 ± 35.7	225.5 ± 38.1	(0.251)
Surgery (10)	192.8 ± 15.6	204.3 ± 24.6	
CPAP + surgery (23)	204.5 ± 17.9	207.6 ± 20.2	
Total	206.7 ± 27.4	214.4 ± 30.6	
Macular inner ring thickness (µm)			F(1.50) = 0.267; *p* = 0.608
CPAP (22)	275.2 ± 18.2	275.6 ± 19.2	(0.005)
Surgery (10)	260.4 ± 12.5	261.8 ± 13.9	
CPAP + surgery (23)	273.8 ± 12.3	274.5 ± 13.3	
Total	271.9 ± 15.8	272.6 ± 16.7	
Macular outer ring thickness (µm)			F(1.50) = 1.438; *p* = 0.236
CPAP (22)	237.5 ± 16.3	237.7 ± 17.9	(0.028)
Surgery (10)	230.7 ± 8	235.2 ± 11.5	
CPAP + surgery (23)	237.8 ± 12.2	238.9 ± 10.1	
Total	236.3 ± 13.6	237.7 ± 13.9	
Macular volume (mm^3^)			F(1.50) = 0.834; *p* = 0.365
CPAP (22)	6.9 ± 0.5	6.8 ± 0.5	(0.016)
Surgery (10)	6.6 ± 0.2	6.8 ± 0.3	
CPAP + surgery (23)	6.9 ± 0.3	6.9 ± 0.3	
Total	6.9 ± 0.4	6.9 ± 0.4	

**Table 5 jcm-11-00815-t005:** OCT findings in OSA patients. RNFL—retinal nerve fiber layer; UAS—upper airway surgery.

	** *n* **	**OCT RETINAL Relevant Findings**
Lin et al. [19]	210	Peripapillary all quadr. RNFL lower in OSA vs. healthy subjects
Sagiv et al. [24]	108	Peripapillary all quadr. RNFL lower in OSA vs. healthy subjects
Casas et al. [20]	96	Only peripapillary nasal RNFL lower in OSA vs. healthy subjects
Adam et al. [21]	43	No differences between OSA and healthy subjects
Kücük et al. [42]	45	No differences between OSA and healthy subjects
Yu et al. [29]	69	Average RNFL thickness lower in severe OSA
Ngoo et al. [43]	44	Peripapillary all quadr. RNFL lower in OSA vs. healthy subjects
Guven et al. [44]	31	Average nasal RNFL thickness lower in severe OSA
Tejero-Garcés et al.	98	Average RNFL thickness lower in severe OSA
	** *n* **	**OCT OPTIC NERVE Relevant Findings**
Lin et al. [19]	210	Peripapillary all quadr. RNFL decreased in OSA vs. healthy subjects
Casas et al. [20]	96	Only peripapillary nasal RNFL decreased in OSA vs. healthy subjects
Kücük et al. [42]	45	No differences between OSA and healthy subjects
Tejero-Garcés et al.	98	Peripapillary atrophy in severe OSA
	** *n* **	**OCT as Monitor of OSA after Treatment**
Zengin et al. [33]	44	After CPAP, all quadr. RNFL decreased compared to controls
Lin et al. [34]	32	After CPAP, inferior and nasal-inf. quadr. RNFL improved compared to controls
Lin et al. [35]	108	After UAS, macular thickness improved compared to controls
Kaya et al. [36]	34	After pharyngoplasty, no significant changes
Naranjo-Bonilla et al. [45]	40	After CPAP, normalization of choroidal thickness
Jayakumar et al. [46]	36	After UAS and CPAP, choroidal thickness and vascularity improved
Tejero-Garcés et al.	98	After UAS and CPAP, foveal thickness and retinal nerve fibers improved in severe OSA compared to controls

## Data Availability

Data are contained within the article.

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
