# Peer review of "Assessment of the Effectiveness of Obstructive Sleep Apnea Treatment Using Optical Coherence Tomography to Evaluate Retinal Findings"

_jcm, 2022, doi:10.3390/jcm11030815_

Round 1
Reviewer 1 Report
The authors have chosen to include both eyes of the subjects in the statistical anaysis, assuming that OSAS influence could be asimmetric, but offer as reference a study of Parkinson disease [32]. But they state that the baseline results did not show such deviation between sides. In ophthalmology it is considered preferably to include only one eye per patient (for instance in glaucoma studies).Perhaps the authors should detail this discussion about including both eyes of a patient and provide other reference about the asymmetry of OSAS influence.
In line with the theory of a neurovascular damage caused by hypoxia, this study has found a reduction of retinal thickness in severe OSAS (compared to the group with mild-moderate OSAS). Were there any eyes in the severe group that had abnormally thin macula compared to the database provided by the OCT (i.e. red sectors)? And were there any quadrants that returned to normal thickness range (i.e. green) after treatment?
The fact that foveal thickness and RNFL thickness improved after treatment in severe cases is considered the most relevant finding of the study. However, the authors did not discuss this finding and did not try to find an explanation. In other degenerative ophthalmic diseases, the idea of regeneration of atrophic retina or SNFR is regarded with suspicion. This matter deserves at least a paragraph in the discussion section of the paper.
Row 285: “Again, this reinforces the need to treat all OSAS patients early to avoid irreversible changes such optic nerve ones.” - Unclear statement, please rephrase
Reviewer 2 Report
Very interesting work. However, some consideration to Authors is due:
1) Introduction and Abstract lack an explanation of the shortcomings of the literature and what new this manuscript brings
2) Number of protocol acceptance
3) How authors calculated the sample size?
4) Did the Authors include patients of all ages or just adults? To specify
5) Authors have to include Maculopathy in exclusion criteria
6) The second aim of the manuscript is the review. The manuscript lacks of material and method of the review
7) If pediatric patients were included, add in the reference
"Bonacci E, Fasolo A, Zaffanello M, Merz T, Brocoli G, Pietrobelli A, Clemente M, De Gregorio A, Longo R, Bosello F, Marchini G, Pedrotti E. Early corneal and optic nerve changes in a paediatric population affected by obstructive sleep apnea syndrome. Int Ophthalmol. 2021 Nov 5. doi: 10.1007/s10792-021-02115-2. Epub ahead of print. PMID: 34738205."
Reviewer 3 Report
- line 44, Repeated sleep fragmentation causes significant oxidative stress with a chronic systemic inflammatory state. Numerous biomarkers have been proposed in this regard in the literature. Cpap or surgical treatment however demonstrated to reduce inflammation biomarkers. please cite doi:10.1016/j.amjoto.2021.103197
- line 42, Intermittent hypoxemia and sleep fragmentation cause continuous sympathetic stimulation with increased cardiac arrhythmogenic risk. please cite doi:10.1016/j.chest.2016.09.014
- line 51, Intermittent hypoxemia disorder also correlated with degeneration of the gray matter of the central nervous system, with increased risk for autonomic pathologies and systemic inflammation. please cite doi:10.3390/jcm10020277
- line 287 Osas demonstrated a significant association with retinal vascular disorerds. Please discuss and cite doi:10.1097/ICU.0000000000000698
- line 41, Cognitive alterations have in particular been associated with both the reduction of the overall performance and the memory abilities of patients with OSA. please cite doi:10.1016/j.jagp.2016.01.134 and https://doi.org/10.339/bs11120180
- line 51, Respiratory indices such as Ahi have recently been associated with the severity of nasal disorders, especially in chronic rhinitic inflammatory states or nasal vascular hyperactivity. please discuss and cite doi:10.3390/medicina56090454
- line 62, A recent trial demonstrated that the treatment of CPAP could improve visual sensitivity and increase retinal thickness in patients with OSA. please discuss and cite doi:10.1016/j.sleep.2019.10.019
- line 71 there is a red typo
- cite the aasm guidelines for diagnostic criteria
- line 108, which type of surgery? when cpap was chosen? please cite doi:10.1111/imj.13606
- line 111 another typo red
- line 117 performed a Muller manovreur?
- sample size was calculated?
- please apply strobe guidelines
- add a flow diagram on study protocol
Reviewer 4 Report
This clinical study investigated the retinal findings in obstructive sleep apnea syndrome (OSAS). A total of 98 eyes from 52 patients (30 eyes) were included in this study. Optic coherence tomography (OCT) was used to monitor the retinal changes during the treatment of OSAS for 6 months. The authors found a substantial improvement in foveal thickness and RNFL average thickness after treatment. The author also found no relationship between changes in retinal parameters and the severity of OSAS. The authors concluded that OCT can evaluate the retina in patients with OSAS and help monitor the results after treatment.
Please provide the data for normal population control, so you can have a comparison.
Round 2
Reviewer 4 Report
my concern is well addressed. thanks